ecology, evolution, genetics

birds, climate, immune genes, major histocompatibility complex, pathogens

**Author for correspondence:**
Emily A. O'Connor
e-mail: emily.o_connor@biol.lu.se

†Joint senior authors.

# Wetter climates select for higher immune gene diversity in resident, but not migratory, songbirds

Emily A. O'Connor, Dennis Hasselquist, Jan-Åke Nilsson, Helena Westerdahl† and Charlie K. Cornwallis†

Department of Biology, Lund University, Lund, Sweden

(iD) EAO, 0000-0001-8702-773X; DH, 0000-0002-0056-6616; J-ÅN, 0000-0001-8982-1064; HW, 0000-0001-7167-9805; CKC, 0000-0003-1308-3995

Pathogen communities can vary substantially between geographical regions due to different environmental conditions. However, little is known about how host immune systems respond to environmental variation across macro-ecological and evolutionary scales. Here, we select 37 species of songbird that inhabit diverse environments, including African and Palaearctic residents and Afro-Palaearctic migrants, to address how climate and habitat have influenced the evolution of key immune genes, the major histocompatibility complex class I (MHC-I). Resident species living in wetter regions, especially in Africa, had higher MHC-I diversity than species living in drier regions, irrespective of the habitats they occupy. By contrast, no relationship was found between MHC-I diversity and precipitation in migrants. Our results suggest that the immune system of birds has evolved greater pathogen recognition in wetter tropical regions. Furthermore, evolving transcontinental migration appears to have enabled species to escape wet, pathogen-rich areas at key periods of the year, relaxing selection for diversity in immune genes and potentially reducing immune system costs.

## 1. Background

The adaptations that allow species to occupy different ecological niches have played a key role in shaping global patterns of biodiversity [1–3]. One crucial factor driving the diversification of ecological niches is pathogens [4,5], as they can have strong effects on host fitness and can vary markedly across time and space due to variation in environmental conditions [6,7]. In particular, climatic factors, such as temperature and rainfall, have a strong impact on local pathogen and vector communities [8–11]. For example, the replication, development and transmission rates of pathogens with life-stages outside of the host and with life-stages in ectotherm vectors are closely linked to ambient temperature [12–14]. Warmer temperatures may promote pathogen richness through faster generation times, higher mutation rates and ultimately increased rates of diversification [15–17]. Similarly, precipitation has been linked to the transmission of water-borne pathogens and pathogens transmitted by vectors whose abundances are linked to rainfall, such as mosquitoes which require stagnant water for breeding [13,14,18]. Despite it being widely recognized that pathogen communities are regulated by climate, it is unclear how specific environmental factors, such as precipitation and temperature, influence the evolution of host immunity. Understanding the immune defences of species occupying different environmental niches is important as it may help explain the emergence of new diseases as well as the global distribution of species, including range expansions and retractions.

Theoretically, pathogen diversity is predicted to drive the evolution of host immune defences that are tailored to the level of disease threat. Species inhabiting warm and wet environments are known to encounter a wider range of pathogens

[16,19–23]. Consequently, hosts in wet tropical areas are expected to evolve defences to cope with greater pathogen diversity than species inhabiting cooler and drier environments. Environmental conditions also vary over the year, but it is unclear whether particular periods of the annual cycle, for instance, the breeding season, are particularly important for host–pathogen interactions, or whether the effects of pathogens accumulate. One way of assessing the importance of annual variation is to compare species that occupy the same area year-round to those that move seasonally between different regions. For example, songbirds include species that are permanent residents in the Palaearctic or Africa, while others migrate seasonally between these two regions. Comparing migrants that breed in the Palaearctic, but winter in Africa, to year-round residents in these regions enables the effects of the environmental conditions during breeding and non-breeding periods on immunity to be disentangled.

A challenge to studying the effects of environmental conditions on the immune systems of resident and migratory species is systematically quantifying host defences across species. A way of overcoming this challenge is to characterize the genetic variation underlying central components of the immune system. In vertebrates, the adaptive immune system plays a crucial role in pathogen defence by mounting a highly specific response to antigenic peptides recognized as non-self. In order for an antigen to be determined as self or non-self, it must be presented to T cells by major histocompatibility complex (MHC) molecules, which are encoded by a set of highly polymorphic genes. Diversity in these genes, both in terms of the number of unique alleles per individual and the sequence divergence in the peptide binding region (PBR) between alleles within individuals (henceforth 'MHC diversity'), increases the number of different antigenic peptides that can be bound and recognized by the immune system [24–28]. However, high MHC diversity may also come at a cost, as it can lead to a higher risk of immunopathology, either through a depletion of the T cell repertoire or by increasing the number of potentially self-reactive T cells [29,30]. The balance between the costs and benefits of high MHC diversity is likely to be modulated by the number of different pathogens hosts encounter. In this way, MHC diversity can provide a window into the historical and ongoing selection from pathogens [24].

Here, we test whether resident and migrant species occupying habitats with warmer and wetter climates have higher MHC class I (MHC-I) gene diversity across the Passerida bird radiation (figure 1). Passerida represent an ideal system for examining the effects of the environment on immunity as there are multiple evolutionary independent colonizations of different environments across the Palaearctic and Africa, as well as multiple origins of migration between these areas [31]. Using this system, we have previously shown that year-round African residents have higher MHC-I diversity than Palaearctic residents and Afro-Palaearctic migrants [31]. However, the ecological factors driving differences in MHC-I diversity across species inhabiting these different biogeographic regions is unknown. We selected target species that encompass the phylogenetic and environmental diversity across Passerida and use a genotyping scheme that allows MHC-I diversity to be quantified across species in a directly comparable manner [31,32]. We intersected species distribution maps with climatic databases and habitat information (proximity to water and lowland versus highland) to test the prediction that species living in wetter and warmer habitats have higher MHC-I diversity than species living in drier and cooler habitats.

## 2. Methods

### (a) Species selection

Thirty-seven species from across the phylogenetic range of the Passerida parvorder were genotyped for MHC-I. The MHC-I data for these species came from two previous studies, with highly comparable genotyping methods [31,32]. The species were chosen to include residents that are present year-round in either the Palaearctic ($N_{species} = 12$) or sub-Saharan Africa ($N_{species} = 15$), as well as Afro-Palaearctic migrants that breed in the Palaearctic but spend their winters in sub-Saharan Africa ($N_{species} = 10$) (see electronic supplementary material, table S1).

### (b) Characterizing the environments occupied by species

#### (i) Climate variables

Climatic time-series data were obtained from high-resolution gridded datasets produced by the Climatic Research Unit (CRU) at the University of East Anglia, UK (CRU TS v. 4.01, https://cru-data.uea.ac.uk/cru/data/hrg [33]). These datasets comprise monthly weather records from 1901 to 2017. We obtained temperature and precipitation data for each species by cross-referencing shapefiles of their distribution ranges, provided by BirdLife International and Handbook of the Birds of the World (http://www.birdlife.org/datazone; accessed April 2017), with the CRU databases. We calculated the median, minimum and maximum temperature and precipitation for each year and then across years (for details, see electronic supplementary material, appendix S1, tables S1 and S2). The minimum and maximum values were extremely highly correlated with the median values for both temperature and precipitation (electronic supplementary material, figure S2). Therefore, we used only the median values in our analyses (electronic supplementary material, table S1). For migrants, calculations were performed separately for their breeding and wintering ranges. Data from May, June and July were used for the breeding ranges, and from November, December and January for the wintering ranges. Exact arrival and leaving dates are not available for many migrants, but the migratory species in our dataset are highly likely to be in their breeding or wintering ranges during these months [34–38]. We also calculated a single value for temperature and precipitation for migrants by combining both sets of seasonal data.

#### (ii) Habitat characteristics

In addition to the specific climate variables, we also characterized each species based on the types of environments they inhabit. We focused on whether or not species tend to live close to water and whether or not they are found in lowland versus highland areas, which influences pathogen communities via its effect on local temperature. Together these characteristics provide information on the finer scale habitat choice of species, which may be missed when using climatic databases from across the entire ranges of species. We characterized the general habitats of species using information collected from the habitat descriptions on the Handbook of Birds of the World Alive website (electronic supplementary material, table S1: data collected on 21–24 August 2018). For details, see electronic supplementary material, appendix S2.

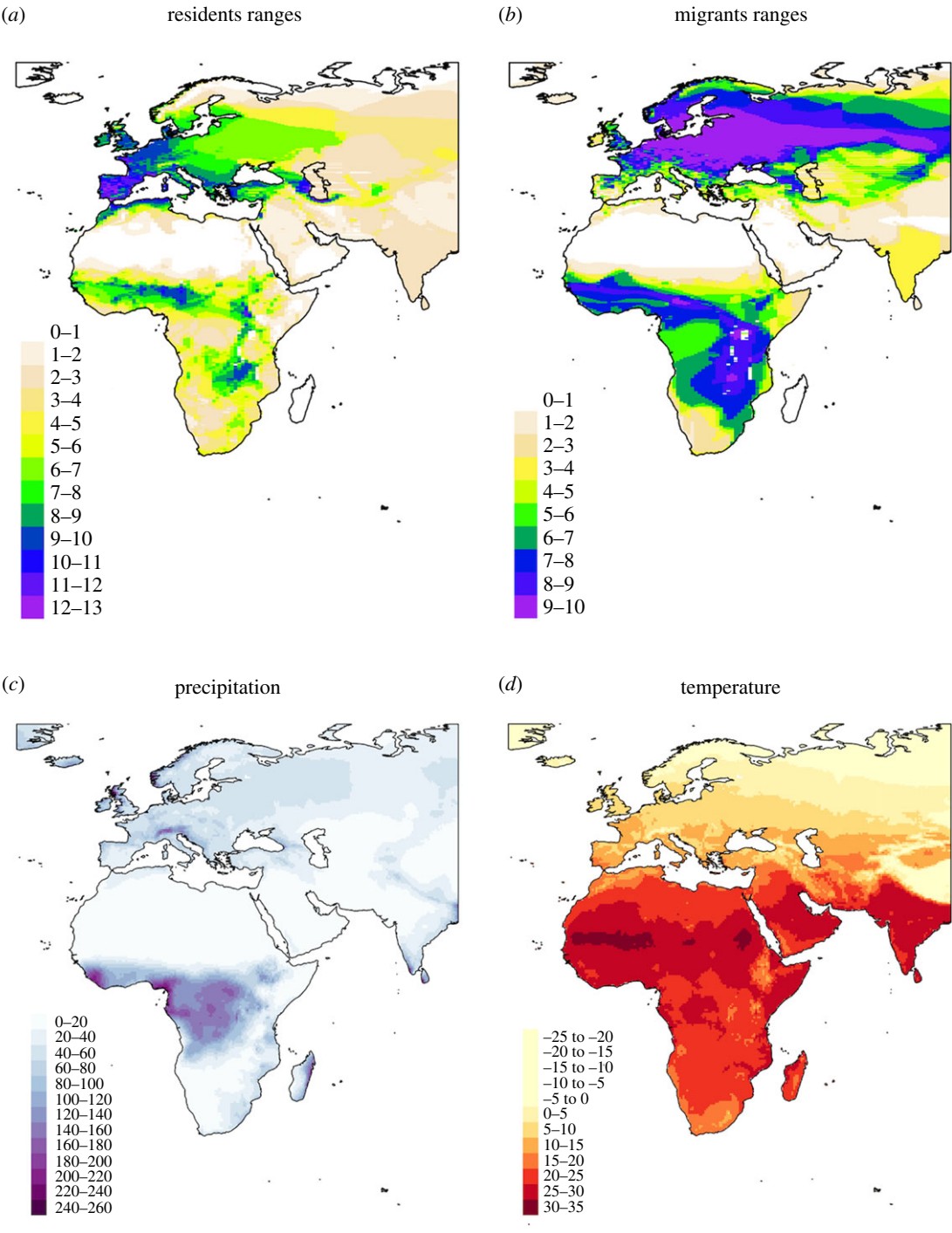

**Figure 1.** Heatmaps of the distribution ranges of the Palaearctic and African resident species in the study (*a*) and the breeding and wintering ranges of the migratory species (*b*). Colours indicate the number of overlapping species. Each point represents a single species. Distribution maps provided by BirdLife international and Handbook of the Birds of the World (2017). Median monthly precipitation (*c*, mm) and temperature (*d*, °C) across the Palaearctic and Africa. The intensity of colour is proportional to the levels of precipitation and temperature. Climate data from monthly weather records (1901–2017) collated by the CRU at the University of East Anglia. (Online version in colour.)

## (c) Estimating major histocompatibility complex class I diversity

We focused on MHC-I genes, which handle antigens from intracellular pathogens such as viruses, certain bacteria, protozoa and fungi. Blood samples from three individuals from each species were used to obtain species-level MHC-I diversity (for some species, one, two or four individuals, see electronic supplementary material, table S1). We have previously demonstrated that this sample size is sufficient for characterizing between species differences in passerines [32]. Estimates of MHC-I diversity for each individual used in the current study can be found in electronic supplementary material, table S3. Full details of the MHC-I genotyping can be found in O'Connor *et al.* [31,32]. In brief,

fragments of exon 3 of MHC-I genes were amplified from DNA extracted from blood. Exon 3 encodes the most variable portion of the PBR providing insight into functionally important diversity. Amplicons from each individual were sequenced using the 454 sequencing platform. The sequence data were used to estimate MHC-I diversity for each species in two ways: (i) the mean number of different MHC-I alleles possessed by individuals and (ii) the mean pairwise sequence divergence (*P*-distance) within the PBR of MHC-I alleles within individuals. The number of different MHC-I alleles within an individual is determined by the number of MHC-I loci they possess and levels of heterozygosity at these loci.

Estimates for the number of MHC-I alleles were available for all 37 species from O'Connor *et al.* [31,32] and estimates of

P-distance for 31 species from O'Connor *et al.* [31] (electronic supplementary material, table S1). For the remaining six species, MHC-I P-distance was calculated using an identical approach to O'Connor *et al.* [31] (electronic supplementary material, table S1). The number of different MHC-I alleles and P-distance represent two distinct estimates of MHC-I diversity as these two variables were not correlated in our dataset (Spearman's rank test calculated using the Hmisc R package [39]: $r = 0.59$, $p = 0.45$, $N_{species} = 37$).

## (d) Estimating demographic effects on major histocompatibility complex class diversity I

It is possible that the diversity of MHC genes is influenced by demographic processes, for example, population bottlenecks and founder events [40,41]. We performed three separate analyses to assess whether demographic processes influenced our estimates of MHC-I diversity. These estimates were conducted on the MHC-I data, as opposed to genome-wide markers, as genes under balancing selection, such as MHC genes, are often affected differently by demographic processes compared to neutral markers [42–44]. Additionally, the MHC region has a pattern of evolution that is distinct from much of the rest of the genome, such as a high evolutionary rate and frequent gene duplications [45,46]. We therefore took a targeted approach and tested for the influence of demographic effects on the genes specifically within the MHC region. The three indicators of demographic effects on MHC-I alleles were haplotype redundancy, the synonymous substitution rate (dS) across the non-PBR sites of MHC-I alleles and the range size for species. For full details and explanation of these analyses, see electronic supplementary material, appendix S3. We included haplotype redundancy, dS and range size in the main statistical models to check whether demographic processes were likely to explain any of the patterns we observed.

## (e) Data analyses

We investigated the relationship between MHC-I diversity and the environments species inhabit using Bayesian phylogenetic mixed models (BPMM) implemented in the R package 'MCMCglmm' [47]. Response variables were either the mean number of different alleles or P-distance within individuals of each species. A Poisson error distribution was used to model the number of alleles and a binomial error distribution was used for modelling P-distance. Climate (temperature or precipitation), habitat-type (living close to water or not, lowland or not), the region species inhabit (Africa or the Palaearctic) and the interaction between temperature and precipitation, as well as precipitation and living close to water, were included as fixed effects. Furthermore, we included dS, haplotype redundancy and range size as covariates in models. Full details of all model specifications and results can be found in electronic supplementary material, tables S4–S11 and S14–S16.

To account for the non-independence of data due to species ancestry, we included a phylogenetic relationship matrix as a random effect in our models. Using the subset tool on the Bird Tree website (http://birdtree.org/), we downloaded a sample of 1500 trees from the posterior distribution of the Hackett all-species backbone tree for the species in our dataset [48]. We ran our models on all 1500 trees to account for uncertainty in phylogenetic relationships, discarding the first 500 trees as a burn-in. For each tree, we ran 10 000 iterations with a burn-in of 9999 and saved the final iteration resulting in a posterior sample of 1000 estimates. Parameter estimates were summarized using the posterior mode (PM) and 95% credible interval (CIs). Terms were considered statistically significant when both the 95% CIs did not span 0 and pMCMC values (the number of iterations greater or less than zero for regressions, or the number of iterations when one level is greater or less than the other level

when comparing groups, divided by the total number of iterations) were below 0.05 [47].

We specified inverse-Wishart priors ($V = 1$, nu = 0.002) for all random effects that led to all models converging. Model convergence was tested by repeating each analysis three times and examining the correspondence between chains in R using the 'coda' package v. 0.16–1 [49] by (i) visually inspecting the traces of the MCMC posterior estimates and their overlap; (ii) calculating the autocorrelation and effective sample size of the posterior distribution of each chain; and (iii) using Gelman and Rubin's convergence diagnostic test, which compares within- and between-chain variance using a potential scale reduction factor [50]. Values higher than 1.1 indicate chains with poor convergence properties. The potential scale reduction factor was less than 1.1 for all the parameter estimates presented.

A potentially confounding factor is latitude as it influences both MHC diversity and climate [31,51]. We tested whether precipitation and temperature were correlated to the latitude of the resident species ranges using similar BPMMs to those described above. The absolute centroid latitude was fitted for each species as a fixed effect and precipitation or temperature as the response variables. We found no significant relationship between latitude and precipitation (posterior mode (PM) = −0.0068, credible intervals (CI) = −0.0211 to 0.0061, pMCMC = 0.14; electronic supplementary material, table S4), but a strong negative association between latitude and temperature (PM = −0.0250, CI = −0.0303 to −0.0187, pMCMC < 0.001; electronic supplementary material, table S4). As latitude and temperature were highly correlated in our dataset, only temperature was included in models to avoid problems of collinearity. We included temperature rather than latitude because when interpreting biological patterns across latitudinal gradients temperature is often invoked as one of the main causal factors [52–54].

## 3. Results

## (a) The evolution of major histocompatibility complex class I in resident species occupying different environmental niches

There was a highly significant relationship between precipitation and MHC-I diversity in resident species, measured as the mean number of different MHC-I alleles per individual (figure 2, PM = 0.0156, CI = 0.0065–0.0227, pMCMC < 0.001; electronic supplementary material, table S5): species living in regions with the highest monthly precipitation had up to eight times more MHC-I alleles than species living in the driest regions. This relationship was particularly strong across African species (PM = 0.0164, CI = 0.0067–0.0241, pMCMC < 0.001), whereas in the Palaearctic, the effect of precipitation on the number of MHC-I alleles was not significantly different from zero (PM = −0.0017, CI = −0.0205 to 0.0325, pMCMC = 0.35). Despite the strong relationship between precipitation and the number of MHC-I alleles, we found no significant association between MHC-I sequence divergence (P-distance) and precipitation in either the African or Palaearctic species (electronic supplementary material, figure S3 and table S5).

Consistent with the results for precipitation, we found that African resident species had more MHC-I alleles when they live close to water (PM = −0.9594, CI = −1.3790 to −0.3454, pMCMC < 0.001; electronic supplementary material, figure S4 and table S6). This difference, however, was no longer significant after precipitation was included in the statistical models (PM = −0.1394, CI = −0.5974 to 0.2947, pMCMC = 0.29; electronic

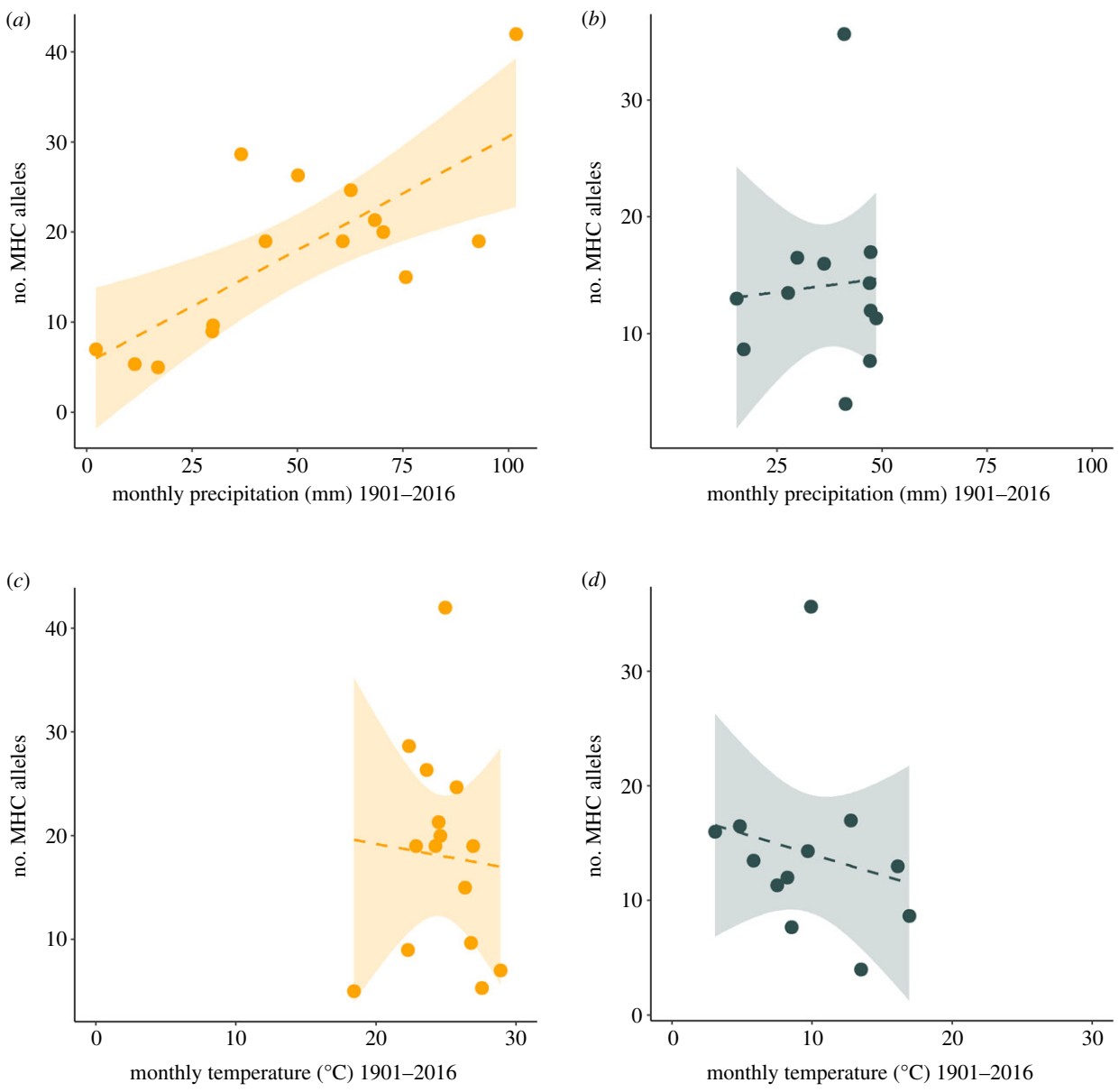

**Figure 2.** Relationship between the median monthly precipitation and the mean number of MHC-I alleles per individual in African residents (*a*) and Palaearctic residents (*b*). Relationship between median monthly temperature and the mean number of MHC-I alleles per individual in African residents (*c*) and Palaearctic residents (*d*). Each point represents a single species. Dashed lines and shaded areas show linear regressions with 95% confidence intervals. (Online version in colour.)

supplementary material, table S6). This suggests that the effect of living near water on MHC-I diversity results from these habitats having higher precipitation. This was supported by the fact that species living close to water also live in areas with higher precipitation (PM = 0.4576, CI = 1.0492–0.0160, pMCMC = 0.03; electronic supplementary material, figure S5 and table S6).

In contrast with precipitation, temperature had little effect on measures of MHC-I diversity (figure 2; number of alleles: PM = −0.0021, CI = −0.0271 to 0.0238, pMCMC = 0.41; electronic supplementary material, table S5; *P*-distance: PM = 0.0081, CI = −0.0099 to 0.0202, pMCMC = 0.21; electronic supplementary material, figure S3 and table S5), and this did not differ between African and Palaearctic species (region × temperature, number of alleles: PM = 0.0221, CI = −0.1359 to 0.1679, pMCMC = 0.34; *P*-distance: PM = −0.0134, CI = −0.1082 to 0.0653, pMCMC = 0.36). We also found that temperature did not modify the effects of precipitation on MHC-I diversity (temperature × precipitation, number of alleles: PM 0.0009, CI −0.0013 to 0.0027, pMCMC = 0.18; *P*-distance: PM = −0.0008, CI = −0.0020 to 0.0007, pMCMC = 0.14; electronic supplementary

material, table S5). It is possible that local variation in temperature experienced by species, for example, due to altitudinal differences, may influence pathogen communities and MHC-I diversity. However, we found no difference between species inhabiting lowland versus highland areas on either measure of MHC-I diversity (electronic supplementary material, table S6). These results indicate that precipitation generates strong selection for MHC-I diversity, at least within Africa, that transcends the effects of the other environmental variables we investigated.

## (b) Testing alternative explanations for the effect of precipitation on major histocompatibility complex I class diversity

It is possible that the effects of precipitation on MHC-I diversity are a by-product of demographic effects on effective population size and genetic drift. However, we found that the relationship between precipitation and number of MHC-I alleles remained significant after controlling for all three

Proc. R. Soc. B 287: 20192675

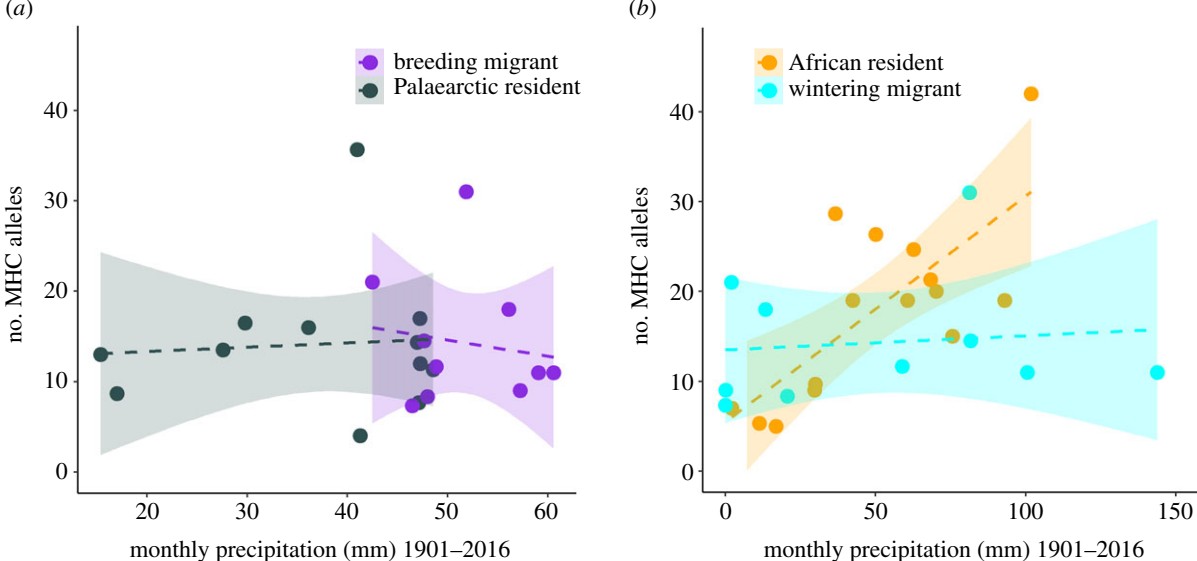

**Figure 3.** Relationship between median monthly precipitation and the number of MHC-I alleles in Palaearctic residents and migrants in their Palaearctic breeding grounds (*a*). Relationship between median monthly precipitation and the number of MHC-I alleles in African residents and migrants in their African wintering grounds (*b*). Each point represents a single species. Dashed lines and shaded areas show linear regressions with 95% confidence intervals. (Online version in colour.)

measures of demographic processes: haplotype redundancy, dS and range size (electronic supplementary material, table S7). We did, however, observe a marginal trend for species with larger ranges to have individuals with more MHC-I alleles (PM = 0.0001, CI = 0.0000 to 0.0003, pMCMC = 0.07; electronic supplementary material, figure S6 and table S7), but this may be due to species with larger ranges encountering more pathogens [55]. The lack of evidence for demographic effects is perhaps not surprising, given that genes under balancing selection, such as MHC genes, have been shown to maintain diversity even through demographic events that reduce genome-wide diversity [42,44,56,57].

The ability to detect a relationship between precipitation and numbers of MHC-I alleles may also be influenced by the extent of variation in rainfall. For instance, the stronger effect of precipitation in African versus Palaearctic residents may be the result of African species experiencing a more extreme range of rainfall (African residents = 2.17–101.8 mm/month, Palaearctic residents = 15.4–48.5 mm/month, figure 2). However, this did not appear to be the case as the association between precipitation and number of MHC-I alleles was still evident when data on African species were restricted to the same range of precipitation as the Palaearctic species, even though this only included five species (electronic supplementary material, figure S7). Furthermore, there was little effect of the change in precipitation across seasons on the number of MHC-I alleles (electronic supplementary material, figure S8 and table S8). Therefore, it is unlikely that the relationship between MHC diversity and precipitation is due to differences in the range of precipitation species experience.

## (c) The evolution of major histocompatibility complex class I in response to environmental niches in migratory species

We investigated the effect of climatic conditions on MHC-I diversity in migrants by examining patterns of precipitation in their breeding and wintering ranges separately. Consistent with the findings from Palaearctic residents, we found little

evidence of an association between MHC-I diversity and precipitation in breeding ranges (figure 3; number of alleles: PM −0.0064, CI −0.0506 to 0.0345, pMCMC = 0.36 PM, *P*-distance: PM −0.0195, CI −0.0494 to 0.0041, pMCMC = 0.05; electronic supplementary material, table S9). However, in contrast with the pattern found in African residents, there was no clear relationship between the number of MHC-I alleles and precipitation experienced by migrants while wintering in Africa (figure 3, PM −0.0025, CI 0.0059–0.0041, pMCMC = 0.31; electronic supplementary material, table S9). Thus, we found a significant difference between African residents and wintering migrants in the relationship between MHC-I and precipitation (PM = 0.0203, CI = 0.0076–0.0272, pMCMC < 0.001; electronic supplementary material, table S10). This difference remained significant even when the African resident species, *Turdus pelios*, was removed as an outlier from the analysis (PM = 0.0149, CI = 0.0044–0.0266, pMCMC = 0.01; electronic supplementary material, table S10). There was no relationship between *P*-distance and precipitation in the migratory species (PM = −0.0024, CI = −0.0048 to 0.0017, pMCMC = 0.17; electronic supplementary material, table S9). There was also no relationship between MHC-I diversity and precipitation across migratory species when combining breeding and wintering range values (electronic supplementary material, figure S9 and table S9), or with temperature and habitat characteristics (electronic supplementary material, tables S9 and S11).

The contrasting results between African residents and migrants suggest that species can escape pathogens by moving away from wet tropical regions when possible. There are, nevertheless, several potential alternative explanations for why the effect of precipitation on the number of MHC-I alleles observed in African residents was lacking in wintering migrants. It could be due to fewer migrant species being included in the study, migrants being in areas that have different amounts of precipitation to residents and/or inaccurate range maps for migratory species. Contrary to this, we found that downsampling the number of African resident species led to the same conclusions (electronic supplementary material, table S12), migrants occupied similar areas with similar environmental conditions to African residents

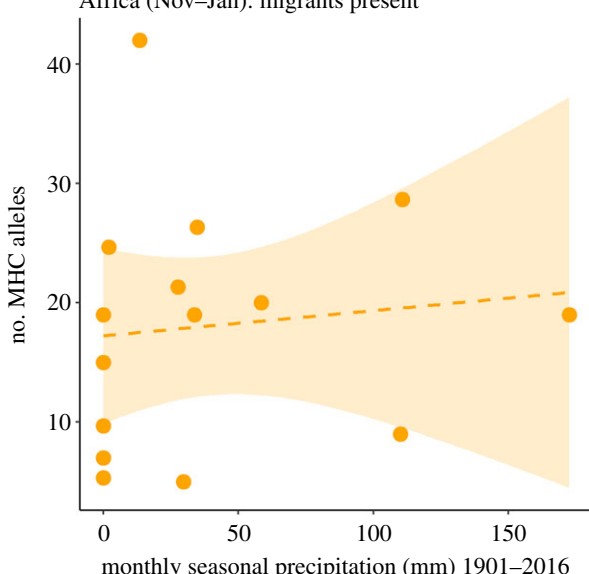

**Figure 4.** Relationship in the African residents between the number of MHC-I alleles and precipitation during the period when the migrants are also present in Africa (November–January). The dashed line and shaded area shows linear regressions with 95% confidence intervals. Each point represents a single species. (Online version in colour.)

(electronic supplementary material, tables S13 and S14), and observational records of migrants from eBird [58] corresponded with range maps (for extended details of these analyses, see electronic supplementary material, appendix S10). This suggests that none of these factors explained the differences between migrants and African residents. However, we did find that migrants may not be in Africa for long enough, or may not be present during periods when selection for higher MHC-I diversity is strongest as the relationship between MHC-I diversity and precipitation in African residents disappeared when data were restricted to the period when migrants coexist with African species (figure 4, PM = 0.0010, CI = −0.0053 to 0.0082, pMCMC = 0.30; electronic supplementary material, table S15). This indicates that prolonged exposure to conditions in Africa may be necessary to drive selection on host immunity (see electronic supplementary material, appendix S10).

## 4. Discussion

Species occupying areas with more diverse pathogen communities are expected to evolve immune responses to cope with the increased threat of disease. Given that pathogen species richness is strongly influenced by environmental conditions and that immunity is costly, host species occupying different environments are predicted to evolve different levels of immune defence [16,51,59]. Here, we provide evidence for this idea by showing that species that inhabit wetter regions, with more pathogens, have higher genetic diversity in key pathogen recognition genes (MHC-I). The effect of precipitation on immune gene diversity was robust to interactions with other environmental factors, but appears to depend on prolonged exposure to wet tropical conditions, as this effect was seen primarily in African residents and not in migrants.

The strong positive association between precipitation and the number of different MHC-I alleles supports the theory that wetter areas have a wider array of pathogens than drier regions. Precipitation is known to influence the species richness of pathogens [16,20], particularly when they have life cycle stages that depend on water, such as water-borne pathogens like avian influenza, or are transmitted by arthropod vectors that are reliant on water, like avian malaria [60,61]. For example, it has been shown that an important predictor of the risk of malaria infection in birds is proximity to a waterbody [62]. This is consistent with the evidence we present here and shows that the increased risk of disease in wetter areas, or when living near water sources, can translate into evolutionary differences across species in the diversity of MHC-I alleles. Beyond the species richness of pathogens, precipitation may also influence the composition of pathogen communities; for example, wet environments may have more similar types of pathogens than wet versus dry environments. An important avenue for future research is investigating whether pathogen communities converge across similar types of environments and, if so, how this shapes immune genes.

The relationship between the number of MHC-I alleles and precipitation was highly significant in African residents, but weaker and non-significant in Palaearctic residents. Africa and the Palaearctic represent two distinct biogeographic regions with very different levels of biodiversity. It is well documented that there is a global latitudinal gradient in biodiversity with higher species richness, including pathogens, in the tropics compared to temperate regions [16,63]. In-line with disease risk being higher in the tropics, we previously found that resident songbirds in Africa have more MHC-I alleles than species in the Palaearctic [31]. The current study provides insights into the environmental factors that may contribute to these differences. The warmer temperatures in Africa combined with higher variation in precipitation may generate more variable and specious pathogen communities compared to the Palaearctic. In turn, greater variation in selection from pathogens across Africa may increase the power to detect the effect of climate on host immune systems. If precipitation does have a more dramatic effect on the species richness of pathogens above a particular temperature threshold, then this may also explain the lack of a linear effect of temperature, or its interaction with precipitation. Ecological tipping points caused by critical temperature changes have recently been discussed [64], but whether or not this is the case for pathogens communities and host–pathogen interactions remains to be established.

While we found a strong association between precipitation and MHC-I diversity in terms of the number of alleles, we observed no such association between precipitation and sequence divergence within the PBR of MHC-I alleles (P-distance). This suggests that having more MHC-I alleles may sufficiently increase the repertoire of antigens recognized in species inhabiting wetter regions. However, in order to understand fully how variation in the amino acid motif of the PBR affects the range of antigens bound by MHC molecules, further work is required to better understand the structure of MHC molecules in birds beyond chickens [65].

Despite the strong effects of precipitation on the MHC-I diversity of African resident species, the MHC-I diversity of migrant species appeared to be independent of climatic conditions. The main reason for this seems to be that prolonged exposure to precipitation in Africa is required to generate selection on MHC-I diversity. Living year-round in Africa is likely to increase the number of different pathogens encountered and promote selection for increased MHC-I diversity. In

this scenario, migrants 'escape' from the broad array of pathogens in tropical Africa, which may have allowed Afro-Palaearctic migrants to lose MHC-I diversity over evolutionary time. This is consistent with our previous finding that Afro-Palaearctic migrants have lower MHC-I diversity than African resident species with which they share a common ancestry [31]. The differences between African residents and migrants in this study also suggest that the region in which these birds breed may be especially formative for immune system evolution [31]. Selection on immune genes may be strongest during the breeding season in both juveniles and adults. For juveniles, it is the period when they first encounter pathogens, and for adults reproducing can be extremely physiologically demanding, increasing their susceptibility to infection [31,66,67].

In a recent comparative analysis of MHC gene copy number variation across birds, Minias *et al.* [68] found that migratory species had more MHC-II gene copies than resident species. Similarly, Whittingham *et al.* [69] reported higher MHC-I gene variation in migratory populations of common yellowthroats, *Geothlypis trichas*, compared to a resident population. However, Whittingham *et al.* [69] also found the opposite patterns in MHC-II genes (i.e. lower variation in migratory populations), and Minias *et al.* [68] reported no association between migration and MHC-I gene copy number. Such differences across studies and species highlight the possibility that MHC-I and MHC-II genes may be subject to different selection pressures, and that escaping from pathogens may apply to some migratory systems, such as Afro-Palaearctic birds, but not others, such as Neotropical migration.

different species that occupy a wide range of environments, with different life-history strategies, it is possible to decouple selection on MHC-I associated with different climatic and habitat factors. The strong positive association between MHC-I diversity and precipitation that we observed in the resident species suggests there is greater selection for MHC-I diversity in wetter areas. Alternatively, this relationship may have arisen because high MHC-I diversity enabled some species to colonize wetter, more pathogen-rich areas. Although the causality behind this association is yet to be determined, understanding the importance of climatic variables for the emergence of disease, and subsequent selection on immunity, is an important step towards predicting the outcome of host–pathogen interactions under different climatic conditions.

**Data accessibility.** All the sequences of the immunity genes included in the study are available on GenBank (accession codes: KU169375–KU169875 and MF477947–MF478976). All other data are either presented within the manuscript or are from publically available sources (climate databases and bird species distribution maps).

**Authors' contributions.** All co-authors contributed to the study design. E.A.O. collected data and performed all analyses. H.W. advised on immune gene analyses and C.K.C. on statistical approaches. E.A.O. wrote the first draft of the manuscript, and all authors contributed substantially to revisions.

**Competing interests.** We declare we have no competing interests.

**Funding.** This project has received funding from: the Knut and Alice Wallenberg Foundation to C.K.C. (grant no. WAF 2013.0129), the Centre for Animal Movement Research financed by a Linnaeus grant (grant no. 349-2007-8690) from the Swedish Research Council and Lund University, the Swedish Research Council (grant no. 2015-05149 to H.W., grant no. 621-2013-4386 to J.-Å.N., grant nos 621-2013-4357 and 2016-04391 to D.H. and grant no. 2017-03880 to C.K.C.) and from the European Research Council (ERC) under the European Union's Horizon 2020 research and innovation programme under grant agreement no. 742646 to D.H. and no. 679799 to H.W.

**Acknowledgements.** We thank two anonymous reviewers whose comments helped improve and clarify this manuscript.

## 5. Conclusion

Our results provide insight into the environmental forces that shape the evolution of vertebrate immune genes. By examining

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
