## [Reviewer comments · Proceedings of the Royal Society B: Biological Sciences]

Review History

RSPB-2019-1706.R0 (Original submission)

Review form: Reviewer 1

Recommendation

Major revision is needed (please make suggestions in comments)

Scientific importance: Is the manuscript an original and important contribution to its field?

Good

General interest: Is the paper of sufficient general interest?

Good

Quality of the paper: Is the overall quality of the paper suitable?

Good

Is the length of the paper justified?

Yes

Should the paper be seen by a specialist statistical reviewer?

No

Do you have any concerns about statistical analyses in this paper? If so, please specify them explicitly in your report.

Yes

It is a condition of publication that authors make their supporting data, code and materials available - either as supplementary material or hosted in an external repository. Please rate, if applicable, the supporting data on the following criteria.

Is it accessible?

No

Is it clear?

No

Is it adequate?

No

Do you have any ethical concerns with this paper?

No

Comments to the Author

In this paper O'Connor et al. examine immune gene variation (MHC class I) in relation to temperature and precipitation in 37 songbird species that inhabit Europe and Africa. They found that resident species in wetter areas, particularly Africa, had higher diversity of alleles at MHC class I than species in drier areas. There was no relationship with precipitation in migrant species. They suggest that the lack of relationship between MHC diversity and environment in migrants is due to the fact that they can escape selection by leaving for part of the year.

This is an interesting explanation, but there are some important papers that are not discussed, some of which could influence the impact and generality of the conclusions.

In particular, the authors do not cite the recent comparative analyses of MHC variation in birds by Minias et al. (2017. *Evolution*; 2019 *Genome Biol Evol*). Minias et al. show that there are different patterns of selection and copy number in passerines and non-passerines. That is not so important in this study because it only included passerines, but the potential for differences should be mentioned. The authors should also mention that Minias et al. also found an effect of migration on copy number at MHC class II, but not class I. All of these differences should be pointed out as there may be important underlying variation that is not being explained.

l. 52 -53. These references are not numbered. Brown et al. 2004 is not in the references. Guernier is #18, Rohde is # 15.

There are a lot of other stylistic changes that need to be fixed in the references. eg, capitalization of article titles (ref 10, 12, 13, 18..).

l. 64. The authors could add some more general references about global pathogen diversity here. There are some more recent human and anuran studies, for example.

l. 93-103. This paragraph (at the end of the Introduction) should tell readers what is new about the study. The authors' previous study in *Nature Ecology & Evolution* examined residents and migrants and inferred parasite diversity from latitude. This study is similar but it uses temperature, precipitation and habitat data, and it also seems to have 10 more species (27 vs 37?). The differences between studies should be made clearer. The relationships with water are definitely new and should be highlighted earlier.

l. 126 & 143. The values for the range (centroid latitude) and environmental variables (temp & precip) --as well as all of the other important data (eg, habitat)-- for each species should also be in Table S1, so all of the main data are in one place and the results can be replicated.

l. 174. change to: "than the variation within species"

l. 202. These are not really independent assays of background genetic variation, because the analyses are still using MHC data. I think the first method (haplotype redundancy) is particularly weak because we do not know the number of loci that these alleles came from.

l. 217-8. "Further details can be found in O'Connor et al. (2018)". If O'Connor et al. (2018) refers to the Nature Ecology and Evolution paper, then it does not really explain the background any more than this paper. Do the authors know, for example, that this measure is correlated with other independent measures of genetic diversity, eg. microsatellite heterozygosity? How about using some other published measure of variation that is truly independent (eg, mtDNA?).

l. 355-6. This conclusion that there were contrasting results between African residents and migrants seems to rest on Fig. 3b, but I do not see any statistics associated with it in the main text. The main text only seems to talk about Fig. 3a. Since this is a major conclusion, the statistical results should be clearly stated in the main text. It is also not clear to me if anything changes if the outlier point in Fig. 3b (the African resident with about 42 alleles) is deleted. How important is this to the conclusions?

l. 433. Speaking of decoupling selection, the authors do not mention the very relevant study by Whittingham et al. (2018, J. Evol. Biol. 31: 1544-57). In that study the authors suggest that different types of selection operate on MHC class I and II. Similar to this study they find that MHC variation was greater in a resident subtropical population of a warbler than a migratory temperate population, but only at class II. The opposite pattern was found at class I (and thus opposite of this study!). This is highly relevant and needs to be mentioned. There is probably not a good explanation for the different results, but they do need to be pointed out.

Decision letter (RSPB-2019-1706.R0)

28-Oct-2019

Dear Dr O'Connor:

I am writing to inform you that your manuscript RSPB-2019-1706 entitled "Wetter climates select for higher immune gene diversity in resident, but not migratory, songbirds" has, in its current form, been rejected for publication in Proceedings B.

This action has been taken on the advice of referees, who have recommended that substantial revisions are necessary. With this in mind we would be happy to consider a resubmission, provided the comments of the referees are fully addressed. However please note that this is not a provisional acceptance.

Sincerely,
Professor Gary Carvalho
mailto:proceedingsb@royalsociety.org

Associate Editor

Board Member: 1

Comments to Author:

Authors analyzed MHC class I diversity for 37 passerines and reported that MHC class I diversity is related to precipitation for the resident birds, but not the migrant bird. The results is intriguing and should be interesting to a broad range of readership. However, as pointed out by the reviewer, the implication of their findings, especially for the migratory species, should be further improved by referring to more recently published literature. Therefore, I would recommend to reject this manuscript for now. But, strongly encourage authors to resubmit their works after carefully revising their manuscript by following the comments of reviewer.

Reviewer(s)' Comments to Author:

Referee: 1

Comments to the Author(s)

In this paper O'Connor et al. examine immune gene variation (MHC class I) in relation to temperature and precipitation in 37 songbird species that inhabit Europe and Africa. They found that resident species in wetter areas, particularly Africa, had higher diversity of alleles at MHC class I than species in drier areas. There was no relationship with precipitation in migrant species. They suggest that the lack of relationship between MHC diversity and environment in migrants is due to the fact that they can escape selection by leaving for part of the year.

This is an interesting explanation, but there are some important papers that are not discussed, some of which could influence the impact and generality of the conclusions.

In particular, the authors do not cite the recent comparative analyses of MHC variation in birds by Minias et al. (2017. *Evolution*; 2019 *Genome Biol Evol*). Minias et al. show that there are different patterns of selection and copy number in passerines and non-passerines. That is not so important in this study because it only included passerines, but the potential for differences should be mentioned. The authors should also mention that Minias et al. also found an effect of migration on copy number at MHC class II, but not class I. All of these differences should be pointed out as there may be important underlying variation that is not being explained.

l. 52 -53. These references are not numbered. Brown et al. 2004 is not in the references. Guernier is #18, Rohde is # 15.

There are a lot of other stylistic changes that need to be fixed in the references. eg, capitalization of article titles (ref 10, 12, 13, 18...).

l. 64. The authors could add some more general references about global pathogen diversity here. There are some more recent human and anuran studies, for example.

l. 93-103. This paragraph (at the end of the Introduction) should tell readers what is new about the study. The authors' previous study in *Nature Ecology & Evolution* examined residents and migrants and inferred parasite diversity from latitude. This study is similar but it uses temperature, precipitation and habitat data, and it also seems to have 10 more species (27 vs 37?). The differences between studies should be made clearer. The relationships with water are definitely new and should be highlighted earlier.

l. 126 & 143. The values for the range (centroid latitude) and environmental variables (temp & precip) --as well as all of the other important data (eg, habitat)-- for each species should also be in Table S1, so all of the main data are in one place and the results can be replicated.

l. 174. change to: "than the variation within species"

l. 202. These are not really independent assays of background genetic variation, because the analyses are still using MHC data. I think the first method (haplotype redundancy) is particularly weak because we do not know the number of loci that these alleles came from.

l. 217-8. "Further details can be found in O'Connor et al. (2018)". If O'Connor et al. (2018) refers to the *Nature Ecology and Evolution* paper, then it does not really explain the background any more than this paper. Do the authors know, for example, that this measure is correlated with other independent measures of genetic diversity, eg. microsatellite heterozygosity? How about using some other published measure of variation that is truly independent (eg, mtDNA?).

l. 355-6. This conclusion that there were contrasting results between African residents and migrants seems to rest on Fig. 3b, but I do not see any statistics associated with it in the main text. The main text only seems to talk about Fig. 3a. Since this is a major conclusion, the statistical results should be clearly stated in the main text. It is also not clear to me if anything changes if the outlier point in Fig. 3b (the African resident with about 42 alleles) is deleted. How important is this to the conclusions?

l. 433. Speaking of decoupling selection, the authors do not mention the very relevant study by Whittingham et al. (2018, *J. Evol. Biol.* 31: 1544-57). In that study the authors suggest that different types of selection operate on MHC class I and II. Similar to this study they find that MHC variation was greater in a resident subtropical population of a warbler than a migratory temperate population, but only at class II. The opposite pattern was found at class I (and thus opposite of this study!). This is highly relevant and needs to be mentioned. There is probably not a good explanation for the different results, but they do need to be pointed out.

Author's Response to Decision Letter for (RSPB-2019-1706.R0)

See Appendix A.

RSPB-2019-2675.R0

Review form: Reviewer 1

Recommendation

Accept as is

Scientific importance: Is the manuscript an original and important contribution to its field?

Excellent

General interest: Is the paper of sufficient general interest?

Good

Quality of the paper: Is the overall quality of the paper suitable?

Excellent

Is the length of the paper justified?

Yes

Should the paper be seen by a specialist statistical reviewer?

No

Do you have any concerns about statistical analyses in this paper? If so, please specify them explicitly in your report.

No

It is a condition of publication that authors make their supporting data, code and materials available - either as supplementary material or hosted in an external repository. Please rate, if applicable, the supporting data on the following criteria.

Is it accessible?

Yes

Is it clear?

Yes

Is it adequate?

Yes

Do you have any ethical concerns with this paper?

No

Comments to the Author

The authors have addressed all of my comments, and I am happy to recommend it for publication.

Review form: Reviewer 2

Recommendation

Accept with minor revision (please list in comments)

Scientific importance: Is the manuscript an original and important contribution to its field?

Good

General interest: Is the paper of sufficient general interest?

Good

Quality of the paper: Is the overall quality of the paper suitable?

Good

Is the length of the paper justified?

Yes

Should the paper be seen by a specialist statistical reviewer?

No

Do you have any concerns about statistical analyses in this paper? If so, please specify them explicitly in your report.

No

It is a condition of publication that authors make their supporting data, code and materials available - either as supplementary material or hosted in an external repository. Please rate, if applicable, the supporting data on the following criteria.

Is it accessible?

Yes

Is it clear?

Yes

Is it adequate?

Yes

Do you have any ethical concerns with this paper?

No

Comments to the Author

I thought this was an interesting study that I enjoyed reading, although it was narrowly focussed, and more could have been done with these data. I have only a few comments / suggestions (in the attached).

Decision letter (RSPB-2019-2675.R0)

02-Dec-2019

Dear Dr O'Connor

I am pleased to inform you that your manuscript RSPB-2019-2675 entitled "Wetter climates select for higher immune gene diversity in resident, but not migratory, songbirds" has been accepted for publication in Proceedings B.

The referee(s) have recommended publication, but also suggest some minor revisions to your manuscript. Therefore, I invite you to respond to the referee(s)' comments and revise your

manuscript. Because the schedule for publication is very tight, it is a condition of publication that you submit the revised version of your manuscript within 7 days. If you do not think you will be able to meet this date please let us know.

[http://datadryad.org/submit?journalID=RSPB&manu=\(Document not available\)](http://datadryad.org/submit?journalID=RSPB&manu=(Document+not+available)) which will take you to your unique entry in the Dryad repository. If you have already submitted your data to dryad you can make any necessary revisions to your dataset by following the above link. Please see <https://royalsociety.org/journals/ethics-policies/data-sharing-mining/> for more details.

Sincerely,
Professor Gary Carvalho
mailto: proceedingsb@royalsociety.org

Associate Editor

Comments to Author:

This version of manuscript had made significant improvements by following the suggestion provided the reviewers in the previous run of submission. It could be accepted for publication if authors could take care two more minor suggestions provided by the reviewer #2.

Reviewer(s)' Comments to Author:

Referee: 1

Comments to the Author(s).

The authors have addressed all of my comments, and I am happy to recommend it for publication.

Referee: 2

Comments to the Author(s).

I thought this was an interesting study that I enjoyed reading, although it was narrowly focused, and more could have been done with these data. I have only a few comments / suggestions (in the attached).

Author's Response to Decision Letter for (RSPB-2019-2675.R0)

See Appendix B.

Decision letter (RSPB-2019-2675.R1)

07-Jan-2020

Dear Dr O'Connor

I am pleased to inform you that your manuscript entitled "Wetter climates select for higher immune gene diversity in resident, but not migratory, songbirds" has been accepted for publication in Proceedings B.

Open Access

Paper charges

Sincerely,

Professor Gary Carvalho

Associate Editor:

Board Member

Comments to Author:

Authors had taken care all the comments made by reviewers. Therefore, I recommend to accept it for publication now.

Appendix A

Rebuttal Letter: Wetter climates select for higher immune gene diversity in resident, but not migratory, songbirds – O'Connor et al.

Proceedings B submission RSPB-2019-1706

We thank the Editor, Professor Gary Carvalho, for the opportunity to revise our manuscript in response to the helpful comments of the Associate Editor and Referee 1. In the pages that follow, we respond to the comments of the Associate Editor and Referee 1 (highlighted in blue), and outline the changes and additions we have made to the manuscript in response to their insightful suggestions. More substantial changes in the revised manuscript are “quoted” and **highlighted in red**. All line numbers refer to the revised text in the ‘track changes’ version of the manuscript.

Associate Editor, Board Member 1

Comments to Author

Authors analyzed MHC class I diversity for 37 passerines and reported that MHC class I diversity is related to precipitation for the resident birds, but not the migrant bird. The results is intriguing and should be interesting to a broad range of readership. However, as pointed out by the reviewer, the implication of their findings, especially for the migratory species, should be further improved by referring to more recently published literature. Therefore, I would recommend to reject this manuscript for now. But, strongly encourage authors to resubmit their works after carefully revising their manuscript by following the comments of reviewer.

Author reply: We thank the Associate Editor for the feedback and encouragement to revise and resubmit our manuscript. We have now addressed each of the comments of Referee 1 by adding citations and discussion of the most recent literature. In addition, we now provided more detailed explanations of our analytical approaches. We have also added extra analyses and provided more data in the Supplementary Material, as suggested by Referee 1.

Referee 1

Comments to Author

In this paper O'Connor et al. examine immune gene variation (MHC class I) in relation to temperature and precipitation in 37 songbird species that inhabit Europe and Africa. They found that resident species in wetter areas, particularly Africa, had higher diversity of alleles at MHC class I than species in drier areas. There was no relationship with precipitation in migrant species. They suggest that the lack of relationship between MHC diversity and environment in migrants is due to the fact that they can escape selection by leaving for part of the year.

This is an interesting explanation, but there are some important papers that are not discussed, some of which could influence the impact and generality of the conclusions.

In particular, the authors do not cite the recent comparative analyses of MHC variation in birds by Minias et al. (2017. *Evolution*; 2019 *Genome Biol Evol*). Minias et al. show that there are different

Rebuttal Letter: Wetter climates select for higher immune gene diversity in resident, but not migratory, songbirds – O'Connor et al.

Proceedings B submission RSPB-2019-1706

patterns of selection and copy number in passerines and non-passerines. That is not so important in this study because it only included passerines, but the potential for differences should be mentioned. The authors should also mention that Minias et al. also found an effect of migration on copy number at MHC class II, but not class I. All of these differences should be pointed out as there may be important underlying variation that is not being explained.

Author reply: We thank Referee 1 for their constructive and considered comments, which have enabled us to substantially improve our manuscript. We agree that it is useful to discuss our findings in light of the highly relevant recent comparative analysis of MHC gene copy number variation in birds by Minias et al (2019). To address this we have included a discussion of how the relationship between MHC genes and migration may differ across different groups of birds and across MHC-I and MHC-II while citing Minias et al 2019. See lines 436-447:

“However, in a recent comparative analysis of MHC gene copy number variation across birds, Minias et al. found that migratory species had more MHC-II gene copies than resident species (87). Similarly, Whittingham et al. reported higher MHC-I gene variation in migratory populations of common yellowthroats, *Geothlypis trichas*, compared to a resident population (88). Nevertheless, Whittingham et al. (88) also found the opposite patterns in MHC-II genes, i.e. lower variation in migratory populations, and Minias et al. (87) reported no association between migration and MHC-I gene copy number. Such differences across studies highlight the possibility that MHC-I and MHC-II genes may be subject to different selection pressures, and that escaping from pathogens may apply to some migratory systems, such as Afro-Palaearctic birds, but not others, such as Neotropical migration. Further studies are required to investigate the sources of variation in the relationship between MHC genes and migration across different groups of birds and between the classes of MHC genes.”

Please note that, as our paper does not include tests of selection, MHC class II genes or non-passerine birds, which is the focus of Minias et al. 2017, we felt it would be too speculative to extend the discussion to include this citation.

l. 52 -53. These references are not numbered. Brown et al. 2004 is not in the references. Guernier is #18, Rohde is # 15.

Author reply: Thank you for spotting this oversight. This is now corrected (l. 50-52).

There are a lot of other stylistic changes that need to be fixed in the references. eg, capitalization of article titles (ref 10, 12, 13, 18...).

Author reply: We have now checked the reference list and corrected any stylistic errors. Please note that as these corrections were done using a reference manager, they are not highlighted in track changes.

l. 64. The authors could add some more general references about global pathogen diversity here. There are some more recent human and anuran studies, for example.

Author reply: We are grateful for this suggestion and have now added the following references: Yang *et al.* (2012), which shows that the outbreak of certain infectious diseases in humans is higher in warmer regions; Nunn *et al.* (2005), which shows that the species richness of primate protozoan parasites increases closer to the equator and Cohen *et al.* (2019), which suggests that a global decline in amphibian species is linked to an interaction between temperature and

Rebuttal Letter: Wetter climates select for higher immune gene diversity in resident, but not migratory, songbirds – O'Connor et al.

Proceedings B submission RSPB-2019-1706

pathogens. See lines 63-64. We were unable to find a specific anuran reference that showed a difference between warm/cool or wet/dry environments in pathogen species richness. If the Referee knows of any such reference, we would happily include it.

l. 93-103. This paragraph (at the end of the Introduction) should tell readers what is new about the study. The authors' previous study in *Nature Ecology & Evolution* examined residents and migrants and inferred parasite diversity from latitude. This study is similar but it uses temperature, precipitation and habitat data, and it also seems to have 10 more species (27 vs 37?). The differences between studies should be made clearer. The relationships with water are definitely new and should be highlighted earlier.

Author reply: We thank Referee 1 for highlighting the need for more clarity over the novelty of our current study compared to O'Connor 2018 *Nat. Ecol. Evol.* In brief, our 2018 paper examined the evolution of MHC-I diversity in relation to the colonization history of Afro-Palaearctic species. This paper indicated that Palaearctic and migrant species evolved from African ancestors that had higher MHC-I diversity, which was subsequently lost as species moved northwards. The 2018 study did not examine any effects of climate or habitat on MHC-I diversity and is therefore clearly distinct from the current study. We have now clarified the differences between the studies in the introduction, which highlights the novelty of our study more directly. See lines 98-101:

“Using this system, we have previously shown that year-round African residents have higher MHC-I diversity than Palaearctic residents and Afro-Palaearctic migrants. However, the ecological factors driving differences in MHC-I diversity across species inhabiting these different biogeographic regions is unknown.”

The MHC data for the 37 species in our current study came from two of our previous studies (O'Connor et al. 2016 & O'Connor et al. 2018). We have now reworded the start of the methods to make this clearer. See lines 111-113:

“Thirty-seven species from across the phylogenetic range of the Passerida parvorder (36) were genotyped for MHC-I. The MHC-I data for these species came from two previous studies, with highly comparable genotyping methods (34,35).”

Later in the methods, there are more details of which MHC-I estimates come from which study (l. 192-197). The study from which each sample came from is also stated in Table S1, referred to at line 197 in the text.

l. 126 & 143. The values for the range (centroid latitude) and environmental variables (temp & precip) --as well as all of the other important data (eg, habitat)-- for each species should also be in Table S1, so all of the main data are in one place and the results can be replicated.

Author reply: This is a good suggestion. We have now added the coordinates of the centroid for each species range (one for each of the breeding and wintering range for migrants) as well as the average temperature and precipitation data for each range in Table S1. The data on the habitat classifications for each species was in Table S2 of the original manuscript. We have now moved this to Table S1 so that all the data analysed can be found in the same worksheet. We also now provide the raw data for the monthly observations of temperature and precipitation for each species, so that all analyses described in our study can be fully replicated using this data in combination with the other data provided in the Supplementary Material.

Rebuttal Letter: Wetter climates select for higher immune gene diversity in resident, but not migratory, songbirds – O'Connor et al.

Proceedings B submission RSPB-2019-1706

l. 174. change to: "than the variation within species"

Author reply: Now corrected (l. 178).

l. 202. These are not really independent assays of background genetic variation, because the analyses are still using MHC data. I think the first method (haplotype redundancy) is particularly weak because we do not know the number of loci that these alleles came from.

Author reply: We previously stated that 'We performed three independent analyses to assess whether demographic processes influenced our estimates of MHC-I diversity'. We now realise that it was not clear that we meant the three analytical approaches were independent of one another, not of the MHC-I genes. To make this clearer we have now reworded this to say "three separate analyses" (l. 205-207).

The aim of including these analyses was to examine the sensitivity of our results to population demographic effects. The MHC region is known for having distinct patterns of evolution as compared to the rest of the genome (i.e. high evolutionary rate, frequent gene duplications etc (Edwards & Hedrick 1998; Piertney & Oliver 2006; Eizaguirre *et al.* 2012; Sorci 2013).

Furthermore, genes under balancing selection, such as MHC genes, are often not affected in the same way as neutral markers by demographic events (Schierup *et al.* 2000; Aguilar *et al.* 2004; van Oosterhout *et al.* 2006). Consequently, we believe it is more appropriate to use the MHC sequence data itself to test for the potential effect of demographic processes. We have explained this in the revised manuscript. See lines 207-213:

"These estimates were conducted on the MHC-I data, as opposed to genome-wide markers, as genes under balancing selection, such as MHC genes, are often affected differently by demographic processes compared to neutral markers (55–57). Additionally, the MHC region has a pattern of evolution that is distinct from much of the rest of the genome, such as a high evolutionary rate and frequent gene duplications (58–61). We therefore took a targeted approach and tested for the influence of demographic effects on the genes within the MHC region."

Concerning the use of haplotype redundancy to estimate the effect of demography on MHC diversity: we are confident that this approach is informative. According to neutral genetic theory, haplotype redundancy (when more than one allele at the nucleotide level codes for the same allele at the amino acid level) will increase with effective population size due to the reduced impact of drift (Hartl & Clark 1997). Not knowing how many loci the alleles come from does not substantially change the theoretical expectations of this pattern. If one considers two different alleles within our dataset: these could represent a single heterozygous locus or a two homozygous loci. The expectation that more redundancy will exist within larger effective populations remains the same for either scenario.

l. 217-8. "Further details can be found in O'Connor et al. (2018)". If O'Connor et al. (2018) refers to the Nature Ecology and Evolution paper, then it does not really explain the background any more than this paper. Do the authors know, for example, that this measure is correlated with other independent measures of genetic diversity, eg. microsatellite heterozygosity? How about using some other published measure of variation that is truly independent (eg, mtDNA?).

Author reply: As MHC genes show different evolutionary patterns to other independent measures of genetic diversity, such as microsatellites (see above for details and references), the estimates are not expected to reflect genome-wide genetic diversity, but rather the effect of demographic processes on MHC-I genes specifically. We are grateful to the Referee for highlighting the need for this clarification. As mentioned above, we have now revised the text to

Rebuttal Letter: Wetter climates select for higher immune gene diversity in resident, but not migratory, songbirds – O'Connor et al.

Proceedings B submission RSPB-2019-1706

further explain and justify our use of these estimates of demographic effects on the MHC-I genes themselves (see above l. 207-213).

l. 355-6. This conclusion that there were contrasting results between African residents and migrants seems to rest on Fig. 3b, but I do not see any statistics associated with it in the main text. The main text only seems to talk about Fig. 3a. Since this is a major conclusion, the statistical results should be clearly stated in the main text. It is also not clear to me if anything changes if the outlier point in Fig. 3b (the African resident with about 42 alleles) is deleted. How important is this to the conclusions?

Author reply: The results of the statistical analyses to support the difference between the migrants and African residents shown in Fig 3b were in Supplementary Table S15 in the original manuscript (Table S9 in the revised version). We now realise that these results should have been clearly stated in the text, which we have corrected in the revised manuscript. We have also included the results of further analyses to demonstrate that removing the African resident with the highest number of MHC-I alleles (*Turdus pelios* with 42 MHC-I alleles) does not alter the result. See lines 355-363:

“However, in contrast to the pattern found in African residents, there was no clear relationship between the number of MHC-I alleles and precipitation experienced by migrants while wintering in Africa (Fig. 3, PM -0.0025, CI 0.0059 to 0.0041, pMCMC = 0.31, Table S8). Thus, we found a significant difference between African residents and wintering migrants in the relationship between MHC-I and precipitation (PM = 0.0203, CI = 0.0076 to 0.0272, pMCMC <0.001, Table S9). This difference remained significant even when the African resident species, *Turdus pelios*, was removed as an outlier from the analysis (PM = 0.0149, CI = 0.0044 to 0.0266, pMCMC = 0.01, Table S9).”

l. 433. Speaking of decoupling selection, the authors do not mention the very relevant study by Whittingham et al. (2018, *J. Evol. Biol.* 31: 1544-57). In that study the authors suggest that different types of selection operate on MHC class I and II. Similar to this study they find that MHC variation was greater in a resident subtropical population of a warbler than a migratory temperate population, but only at class II. The opposite pattern was found at class I (and thus opposite of this study!). This is highly relevant and needs to be mentioned. There is probably not a good explanation for the different results, but they do need to be pointed out.

Author reply: We are grateful to the Referee for the suggestion. Originally, we did not refer to Whittingham et al 2018 because our study examines variation across species, as opposed to MHC variation between populations with a species. However, we agree that the Whittingham et al. (2018) very nicely demonstrates the potential for selection on MHC-I and MHC-II to differ, as well as providing another example of a case where migration may be associated with lower MHC diversity (albeit MHC-II). We now cite Whittingham et al (2018) when discussing the relationship between migration and MHC genes. See lines: 438-447:

“Similarly, Whittingham et al. reported higher MHC-I gene variation in migratory populations of common yellowthroats, *Geothlypis trichas*, compared to a resident population (88). Nevertheless, Whittingham et al. (88) also found the opposite patterns in MHC-II genes, i.e. lower variation in

Rebuttal Letter: Wetter climates select for higher immune gene diversity in resident, but not migratory, songbirds – O'Connor et al.

Proceedings B submission RSPB-2019-1706

migratory populations, and Minias et al. (87) reported no association between migration and MHC-I gene copy number. Such differences across studies highlight the possibility that MHC-I and MHC-II genes may be subject to different selection pressures, and that escaping from pathogens may apply to some migratory systems, such as Afro-Palaearctic birds, but not others, such as Neotropical migration. Further studies are required to investigate the sources of variation in the relationship between MHC genes and migration across different groups of birds and between the classes of MHC genes.”

REFERENCES

- Aguilar, A., Roemer, G., Debenham, S., Binns, M., Garcelon, D. & Wayne, R.K. (2004). High MHC diversity maintained by balancing selection in an otherwise genetically monomorphic mammal. *Proc. Natl. Acad. Sci.*, 101, 3490–3494.
- Cohen, J.M., Civitello, D.J., Venesky, M.D., McMahon, T.A. & Rohr, J.R. (2019). An interaction between climate change and infectious disease drove widespread amphibian declines. *Glob. Chang. Biol.*, 25, 927–937.
- Edwards, S. V. & Hedrick, P.W. (1998). Evolution and ecology of MHC molecules: From genomes to sexual selection. *Trends Ecol Evol*, 13, 305–311.
- Eizaguirre, C., Lenz, T.L., Kalbe, M. & Milinski, M. (2012). Rapid and adaptive evolution of MHC genes under parasite selection in experimental vertebrate populations. *Nat. Commun.*, 3, 621–626.
- Hartl, D.N. & Clark, A.G. (1997). *Principles of Population Genetics*. 4th edn. Sinauer Associates, Sunderland, MA.
- Nunn, C.L., Altizer, S.M., Sechrest, W. & Cunningham, A.A. (2005). Latitudinal gradients of parasite species richness in primates. *Divers. Distrib.*, 11, 249–256.
- van Oosterhout, C., Joyce, D.A., Cummings, S.M., Blais, J., Barson, N.J., Ramnarine, I.W., et al. (2006). Balancing selection, random genetic drift, and genetic variation at the major histocompatibility complex in two wild populations of guppies (*Poecilia reticulata*). *Evolution*, 60, 2562–2574.
- Piertney, S.B. & Oliver, M.K. (2006). The evolutionary ecology of the major histocompatibility complex. *Heredity (Edinb.)*, 96, 7–21.
- Schierup, M.H., Vekemans, X. & Charlesworth, D. (2000). The effect of subdivision on variation at multi-allelic loci under balancing selection. *Genet. Res.*, 76, 51–62.
- Sorci, G. (2013). Immunity, resistance and tolerance in bird-parasite interactions. *Parasite Immunol.*, 35, 350–361.
- Yang, K., LeJeune, J., Alsdorf, D., Lu, B., Shum, C.K. & Liang, S. (2012). Global distribution of outbreaks of water-associated infectious diseases. *PLoS Negl. Trop. Dis.*, 6.

Appendix B

Response to Referees: Wetter climates select for higher immune gene diversity in resident, but not migratory, songbirds – O'Connor et al.

Proceedings B submission RSPB-2019-2675

We thank the Editor, Professor Gary Carvalho, for accepting our manuscript subject to minor revisions. We appreciate the helpful feedback of the Associate Editor and both Referees #1 and #2. Below, we respond to the comments of the Associate Editor and Referee 1 and 2 (highlighted in blue), and outline the changes we have made to the manuscript in response to the suggestions of Referee #2. Changes to the main text of the manuscript are “quoted” and **highlighted in red**. All line numbers refer to the revised version of the manuscript.

Associate Editor

Comments to Author

This version of manuscript had made significant improvements by following the suggestion provided the reviewers in the previous run of submission. It could be accepted for publication if authors could take care two more minor suggestions provided by the reviewer #2.

Author reply: We are pleased that the Associate Editor is satisfied with the changes we made during the last revision. We have now addressed the two minor suggestions of Referee #2, by making the data they requested available in the supplementary material and adding a discussion of future research questions.

Referee 1

Comments to Author

The authors have addressed all of my comments, and I am happy to recommend it for publication.

Author reply: We thank Referee #1 for their help in substantially improving our manuscript.

Referee 2

Comments to Author

I thought this was an interesting study that I enjoyed reading, although it was narrowly focussed, and more could have been done with these data. I have only a few comments / suggestions (in the attached).

Attached comments: This study examines the MHC class I variation in 37 species of songbird that inhabit diverse environments, including African and Palearctic, and finds a strong positive association between MHC-I diversity and precipitation in resident species but not the migrating species.

They argue that this may be the result of greater selection for MHC-I diversity in wetter areas, acknowledging the alternative interpretation, i.e. this relationship may have arisen because high MHC-I diversity enabled some species to colonise wetter, more pathogen rich areas. They have taken confounding factors into account such as demographic differences, using three different approaches to estimate the impact of genetic drift. Their results are remarkable, and contrast those of other studies on bird MHC class II (which showed that migratory species had more MHC-II gene copies than residents), and

Response to Referees: Wetter climates select for higher immune gene diversity in resident, but not migratory, songbirds – O'Connor et al.

Proceedings B submission RSPB-2019-2675

another study on MHC class I in the common yellowthroat (which also showed higher MHC class I diversity in the migratory populations of this species). This literature has been cited, and these contrasting findings demonstrate the complexity of MHC biology. I thought this was an interesting study (albeit narrowly focussed), and I have only a few comments / suggestions.

Author reply: We are grateful to Referee #2 for their encouraging and positive feedback on our manuscript.

Minor comments:

1) "Blood samples from three individuals from each species were used to obtain species-level MHC-I diversity (for some species one, two or four individuals, see Table S1). We have previously demonstrated that this sample size results in similar estimates of MHC-I diversity at a species level compared to those from studies using many more individuals, and that between species variation in these estimates of MHC-I diversity is much greater than the variation within species..."

Can you please show the number of alleles per individual? Unfortunately, the Appendix was not included in the proofs or on the ScholarOne, so perhaps these data were given. If not, please add because with some species being presented by just 1 or 2 individuals, knowing the between-individual variation is important. For example, if one of your individuals was relatively inbred, this could significantly reduce the number of alleles.

Author reply: This is an excellent suggestion. We have now added a table to the supplementary material providing MHC-I diversity estimates for each individual in the dataset (Table S3). We refer to this data in the text (lines 132 to 133) as:

"Estimates of MHC-I diversity for each individual used in the current study can be found in Table S3."

2) The MS is focussed on testing a single hypothesis, but so much more can be done with these data. For example, I would like to know whether the PBR motives of the relatively small number of alleles of species in dry areas cover the same epitope space as those of wet areas. An analysis of supertype variation could be very revealing.

Author reply: We agree with Referee #2 that the question of whether the epitope space dealt with by MHC-I alleles of species in dry versus wet areas differs is intriguing, and one we have previously considered. However, the analyses required to address this question accurately, such as calculating reliable MHC superotypes or in silico predictions of the antigen binding properties of different MHC alleles, require more data than is currently available. For example, better knowledge of the structure of MHC molecules in birds, beyond chickens, is needed before accurate predictions can be made on how variation in the PBR motif impacts the actual epitope space handled by MHC molecules. As a result, the validity of supertype analyses in passerine birds is unclear and for these reasons we decided not to include them. We have now edited the manuscript to acknowledge this as an important area of research that requires further development (lines 350-356):

Response to Referees: Wetter climates select for higher immune gene diversity in resident, but not migratory, songbirds – O'Connor et al.

Proceedings B submission RSPB-2019-2675

“Whilst we found a strong association between precipitation and MHC-I diversity in terms of the number of alleles, we observed no such association between precipitation and sequence divergence within the PBR of MHC-I alleles (P-distance). This suggests that having more MHC-I alleles may sufficiently increase the repertoire of antigens recognised in species inhabiting wetter regions. However, in order to understand fully how variation in the amino acid motif of the PBR affects the range of antigens bound by MHC molecules, further work is required to better understand the structure of MHC molecules in birds beyond chickens (65).”

Similarly, is the mean pairwise sequence divergence (P-distance) within the PBR of two species of “wet” birds smaller than that between a “dry” and a “wet” bird? (Or between a migrant and a resident, etc.?). There are many other things that can be done with these data, but I understand that this would go beyond the scope of this paper. (It would be nice to examine these things though!).

Author reply: We appreciate the referee’s enthusiasm for the potential of our data. We agree that the data could be used to address other questions, beyond those described in the current manuscript. Our main question was to test whether climatic factors and habitat explain MHC diversity in resident and migrant species across the Afro-Palaearctic region. We included analyses directly relevant to testing this idea, which comprised of 35 analyses, 4 main figures and 11 supplementary figures. We note that Referee #2 was unable to access the supplementary material during their review, so are perhaps unaware of the breadth of the analyses.

Examining the differences in alleles across species within and between different environments (e.g. wet-wet vs wet-dry) and between different life-history strategies (e.g. migrant vs resident), as raised by the referee, would be interesting. However, as the referee also points out, this is beyond the scope of this manuscript, given that this is not directly related to our original question and the extent of the analyses already presented. However, we have now edited the manuscript to highlight that this would be an important future avenue of research (lines 327-331):

“Beyond the species richness of pathogens, precipitation may also influence the composition of pathogen communities, for example, if two wet environments have more similar pathogens than a wet versus a dry environment. An important avenue for future research is investigating whether pathogen communities converge across similar types of environments and, if so, how this shapes immune genes.”